# Honey-Propolis-Engineered Collagen Peptides as Promising Wound-Healing Matrix in Mouse Model

**DOI:** 10.3390/molecules27207090

**Published:** 2022-10-20

**Authors:** Hairul-Islam Mohamed Ibrahim, Muthukumar Thangavelu, Ashraf Khalifa

**Affiliations:** 1Biological Science Department, College of Science, King Faisal University, P.O. Box 400, Al-Ahsa 31982, Saudi Arabia; 2Molecular Biology Division, Pondicherry Centre for Biological Sciences and Educational Trust, Pondicherry 605004, India; 3Department BIN Convergence Tech, Jeonbuk National University, 567 Baekje-dearo, Deokjin, Jeonju 54896, Jeollabuk-do, Korea; 4Department PolymerNano Sci and Tech, Jeonbuk National University, 567 Baekje-dearo, Deokjin, Jeonju 54896, Jeollabuk-do, Korea; 5Botany and Microbiology Department, Faculty of Science, Beni-Suef University, Beni-Suef 62511, Egypt

**Keywords:** collagen, propolis, skin, wound progression, VEGF

## Abstract

In this study, collagen hydrolysates (CHDs) were fabricated with honey-propolis wax (HPW), structurally modified as a sponge matrix, and experimentalized on wound healing in a mouse model. The scaffold was characterized by means of in vitro enzymatic degradation; in vitro HPW release; and in vivo wound-healing mouse model, wound-healing-specific RNA, transcripts, and protein markers. The functional activity of the HPW extracted from raw propolis was determined using total flavonoids, antioxidant scavenging assays, and anti-hemolytic principles. The results indicated that HPW had a high flavonoid content (20 μg/mL of wax) and antioxidant activities. The effective concentration (EC50) of HPW was estimated (28 mg/mL) and was then used in the subsequent in vivo experiments. Additionally, the dopped mixture of CHDs and HPW substantially enhanced the wound-healing process and regulated wound biochemical markers such as hexoseamine and melondialdehyde. CHDs- HPW upregulated the expression of growth factors including vascular endothelial growth factor (VEGF) (2.3-fold), fibroblast growth factor (FGF) and epidermal growth factor (EGF) (1.7-fold), and transforming growth factor-beta (TGF-β) (3.1-fold), indicating their potential capacity to perform wound re-epithelialization and the loading of ground tissue. Pro-inflammatory markers IL-1 β (51 pg/mL) and TNF-α (220 pg/mL) were significantly reduced in the CHD-HPW-treated wound. These interesting results were further confirmed using mRNA and protein growth factors from the wound, which enhanced the load of collagen-I in the wound site. In conclusion, CHDs-HPW exhibited a significant reduction in inflammation and inflammatory markers and helped to obtain a faster wound-healing process in a mouse model. The newly engineered biosponge could be developed as a promising therapeutic approach for the regeneration and repair of damaged human skin in the future.

## 1. Introduction

The skin is the largest organ of the human body that protects as well as acts as a barrier against mechanical, environmental, internal, external, and microbial damage [1,2]. Skin damage and wound healing are also major issues being faced by medical professionals, irrespective of the type of wounds [3]. Worldwide, more than six million people struggle with different kinds of wounds, such as vascular ulcers and pressure injuries, that change their entire lifestyle and bring down their quality of life [4]. Acute wounds heal faster, with the successful restoration of tissues, whereas chronic wounds are severe and take more than 11–12 weeks to heal; sometimes, the process fails, leading to the incomplete restoration of skin structures [5]. Therefore, there is a major need for the design of a cost-effective wound dressing material to enhance the wound-healing process and to restore the structure and the function of the skin. Therefore, there is a critical need for the development of a low-cost wound dressing material to improve wound healing and restore skin structure and function.

There are numerous treatments available on the market for wound and burn management, the majority of which are wound dressings. They are made of a wide variety of material types with different claims that vary depending on the type of wounds they are used for. The main concern, however, is how well they alleviate the healing process [6]. A proper wound dressing should provide a favorable microenvironment for wound healing, such as proper gas permeability, germ and dust permeability, low adhesion to the wound site, flexibility, durability, water loss prevention, hemocompatibility, and ease of separation from wound surface during dressing [7,8,9]. The type of wound dressing and the materials used to prepare wound dressing play an important role in the wound-healing process. Several studies highlighted the importance of using traditional medicines in wound dressings to improve the wound-healing process, also due to their therapeutic properties, low cost, bioavailability, and efficacy [8,10].

The wound dressing material developed in this study was made out of collagen hydrolysates, which form a biodegradable natural tissue matrix that provides skin strength, aids in wound healing and tissue regeneration, is little antigenic, is biocompatible, has the ability to promote cell attachment and proliferation, and is non-toxic. Collagen has well-documented structural, chemical, and physical properties [8,11], and when combined with propolis, a sticky substance formed by the resin collected from trees by bees and secreted by their maxillary glands, it forms a sticky substance. It is made out of 50% of balsam resin, 30% of wax, 10% of essential and aromatic oils, 5% of pollen, and 5% of other ingredients, such as wood fragments [12]. It was recently reported that an innovative wound-healing dressing based on a propolis (EPP-AF^®^)-containing self-microemulsifying formulation incorporated in biocellulose membranes was developed and showed significant good results in preclinical trials [13,14].

Propolis is believed to have antibacterial, anti-inflammatory, antiseptic, antimycotic, anesthetic, spasmolytic, antifungal, antiulcer, anticancer, and immunomodulatory properties [13,15], as well as wound-healing properties. It can decrease scar formation, shorten healing time, boost wound contraction, accelerate tissue repair, and ultimately improve patients’ quality of life in skin wound healing [16]. More than 300 different compounds were characterized so far, including aliphatic acid, fatty acids, ketones, terpenoids, vitamins, etc. [13,15,17]. A considerable interest in the effects of propolis on animal wound healing can be observed in clinical and experimental settings [18,19]. The evaluation of collagen scaffolds for coupling biomolecules for treated wound dressing material in mice was studied. Collagen peptides were used as a wound dressing material in the form of powder, amorphous gels/pastes, gel-impregnated dressings, and spongy pads. This material mimics a natural protein molecule, is involved in host enzymatic degradation, and liberates nano-sized bioactive peptides with differential biological functions [20]. It was already reported for its anti-inflammatory [21], antibacterial, [20], and antioxidative effects [21], alongside other effects. In topical applications, the collagen material transmits a false signal to keratinocyte-based fibroblast cells to synthesize new collagen fibers; moreover, it possesses chemotactic properties, as shown by its ability to regulate cell migration and proliferation in the wound microenvironment [22].

However, little is known about the potential roles of honey-propolis-engineered collagen peptides as a strategy for wound healing. Therefore, in the current study, collagen sponge matrixes (CHDs) with and without propolis (PE) were evaluated for their physiochemical properties, drug-release patterns, in vitro cell migration assay using the HACAT cell line, in vivo wound healing in animals, and their effect on growth factor expressions and various biochemical parameters.

## 2. Results

### 2.1. Evaluation of Total Flavonoids from Extracted Propolis Wax from Bee Hive

A hydroalcoholic solvent was used to extract flavonoid-rich propolis from raw propolis wax. The extract of propolis wax was rich in flavonoids. The hydroalcoholic extract showed 38 μg/mg of flavonoid content in the total extract. This extract was further used to evaluate wound-healing applications. About 8 g of flavonoid-rich extract, called honey-propolis wax (HPW), was obtained from 100 g of raw propolis. The content was correlated with the quercetin internal control.

### 2.2. Antioxidant Activity of HPW

DPPH was used to determine the scavenging ability of radicals in different concentrations of HPW. The antioxidant activity was noted, and the EC50 of HPW was considered as a 28 mg/mL concentration. It was further tested for FRAP and hemolytic activity. The Fe reducing power was significant in HPW at concentrations of 20 mg/mL (42%) and 50 mg/mL (60%).

To study the cytoprotective activity of HPW under oxidative stress conditions, in vitro mouse RBC-lysis assays were performed. As presented in Figure 1, HPW at a concentration 50 mg/mL showed 17% of hemolysis, which was considered as representing low biocompatibility. Concentrations > 50 mg/mL indicated biocompatibility and cytoprotective agent function. These concentration values did not significantly damage the RBCs. Based on this assay, the HPW EC50 concentration for hemolysis was a concentration < 100 mg/mL (Figure 1D). The rate of RBC lysis was compared with 10% triton x-100 agent. It showed significant hemolysis and was noted as 100%. Altogether, the concentration for further analyses chosen was the 28 mg/mL concentration for collagen hydrolysate sponge studies.

### 2.3. Analysis of Biosponge Fabricated with HPW

#### FTIR Spectroscopy

The FTIR spectrum is a structural analysis tool for studying collagens and the configuration of polypeptide chains. Collagen peptides formed a unique right-handed triple superhelical rod consisting of three almost identical polypeptide chains. The amide group of the peptides dehydrated to form a polymer mixture with the natural products (Figure 2A). The FTIR signals of HPW were observed at 1226, 1174, and 1133 cm^−1^. The CHD sponge showed the asymmetric CO-O-CO stretching belonging to the class of anhydrides at 1159 cm^−1^ and the antisymmetric bending of the methyl groups in acetate at 1374 and 1430 cm^−1^. The shift in the spectral wavelength in the combination factor showed modified peaks at 2433, 2325, and 1240 cm^−1^.

The biological stability of the collagen sponge was assessed via in vitro collagenase activity. The in vitro enzymatic degradation of the bilayer material is shown in Figure 2B. The collagen peptide sponge underwent controlled degradation upon the secretion of lytic lysosomal enzymes in the wound site. The weight loss of CHDs-HPW due to enzymatic degradation was significantly high due to the degradation of collagen peptides operated by proteolytic enzymes (Figure 2B). These observations are relative to the dressing material preparation and periodical redressing intervals and concluded after 3 days; regarding the material stability, they could be altered within 4 days, as CHDs-HPW showed 56.3% of degradation and CHDs alone showed 45.5%. This material could be a suitable dressing material for wound-healing applications.

The in vitro drug-release study of the HPW-fabricated collagen peptides is shown in Figure 2C. The CHDs loaded with HPW were used to reach the absorption state in the wounded site at the desired time point. The in vitro HPW release started at 14.2%, and the controlled experiments lasted 4 h.

However, the decrease in the initial burst release corresponded to the drug that partially bounded to the hydrophobic matrix bonding. Nevertheless, the sustained drug-release behavior suggested the prevention of infection at the wound site. However, the release of HPW in the wound site and the degradation of collagen peptides increased the absorption and re-epithelization of keratinocytes. Usually, several mathematical models can be used to define the dissolution. The drug-release kinetic profile could be correlated with some mathematical models and is represented in Figure 2C. Based on the regression coefficient, the release of HPW from the sponge matrix exhibited a direct, nearly constant drug release.

### 2.4. In Vivo Evaluation of Collagen Peptides (CHDs) Fabricated with HPW

Macroscopic observations and planimetric analyses were carried out on all tested mice, where one mouse represented each group, as shown in Figure 3A. Epithelization and wound recovery were noted every 10 days up to 2 weeks. The photographic evaluation of CHD and CHD-HPW groups exhibited significant wound recovery in each of the tested groups and found that the synergistic combination of CHDs-HPW showed higher wound recovery and re-epithelization than the HPW-only group (Figure 3B). With CHDs-HPW, wound healing was significantly improved compared with CHD alone (Figure 3A,B). The collagen peptides manipulated with HPW brought relayed and sustained release. HPW supported the growth of skin cells and differentiation, as shown in Figure 4. The visual evidence of in vivo wound healing was obtained by taking photographs using a 4K digital camera from a uniform distance with fixed zoom in all mouse experiments. The results indicated faster healing in the experimental CHD-HPW group than in the other tested groups.

The results of the percentage of wound contraction, total proteins, hexoseamine, and lipid peroxidation are presented in Figure 4A–D. The planimetric evaluation displayed the measurement area of wound size in mice. In the present study, HPW with CHDs exhibited a significant rate of wound closure from the 10th day of the post-wound period (Figure 4A). The percentage of the wound area contraction in the HPW-CHD group was nearly 50% on the 10th day of treatment, which was highly significant compared with the other two groups (Figure 4A). Similarly, the total proteins content in excised wounds was significantly high in the HPW-CHD group (Figure 4B). The levels of granulation markers such as hexoseamine and MDA were significantly low and high, respectively, in excised wound tissue from the treated group, HPW-CHD, compared with the untreated control one (Figure 4C,D).

The production of cytokine proteins (VEGF, EGF, TNF-alpha, IL-1beta, and TGF-Beta) in excised wound tissue from mice is shown in Figure 5A–E. The expression levels of VEGF, EGF, and TGF-Beta were substantially increased in the CHD-HPW group on the 20 th day of the post-wound period. After wound induction, these three growth factors were stimulated by collagen peptides incorporated with HPE (Figure 5A–C). The production of TNF-alpha (220 pg/mL) and IL-1beta (51 pg/mL) was reduced in the treated group from the 10th day onward.

The findings of wound-healing growth factor mRNA and protein expression in collagen-sponge-based HPW-treated excised wound tissue are presented in Figure 6. The gene expression of VEGF (2.3-fold), EGF (1.7-fold), and TGF (3.1-fold) was progressively regulated in the CHD-HPW group (Figure 6A). The expression of collagen-I and VEGF protein markers was significantly increased in the wound site in the CHD-HPW and CHD groups, indicating their potential capacity to perform wound re-epithelialization and the loading of ground tissue (Figure 6B,C). These observations clearly indicated that the collagen peptides engineered using HPW attenuated the wound severity and improved the healing process in the mouse model.

## 3. Discussion

Propolis is a resinous mixture produced by honey bees and contains high amounts phenols and flavonoids [23]. It has a wide range of biological activities, including antimicrobial, anti-inflammatory, antiseptic, antimitotic, anesthetic, spasmolytic, antiulcer, anticancer, and immunomodulatory properties, as well as wound-healing properties [24,25,26].

In this study, collagen hydrolysates (CHDs) were fabricated with honey-propolis wax (HPW) and structurally modified as a sponge matrix; then, they were tested for wound healing in a mouse model. To characterize the scaffold, we used in vitro enzymatic degradation; in vitro HPW release; and in vivo wound-healing mouse model, wound-healing-specific RNA transcripts, and protein markers. Total flavonoids, antioxidant scavenging assays, and anti-hemolytic principles were used to determine the functional activity of the HPW extracted from raw propolis.

The extraction yield of propolis wax from a raw bee hive was much lower (14.1%) than those (35%) reported in other studies [27,28]. Such difference could be attributed to the honey type, the extraction method and conditions applied, and the physicochemical and geographical position of the hive [28,29]. The flavonoid content in HPW was significantly higher than those previously reported [30,31], indicating the high quality of the yield. In this study, the antioxidant activity of HPW was significantly high, as evidenced by the reducing power of DPPH and Fe ion chelating activities. Available reports indicated that flavonoids display versatile biological activities, including binding affinity against polymers, the chelation of metal ions, the catalysis of electron transport, and scavenging oxidative free radicals [32,33]

Sponge collagen peptides were digested at the wound site via the secretion of lytic lysosomal enzymes. The high weight loss of CHDs-HPW was most likely due to the activity of proteolytic enzymes. The degradation rate of the sponge matrix acted upon the biological stability of HPW and its suitability as a dressing material for wound-healing applications. Furthermore, HPW improved cell proliferation and differentiation. Similar results were reported by [34]. In this study, about 30% of HPW release occurred within 24 h and attained 78% after 60 h, indicating a faster release of collagen in the treated group than that in the untreated groups. The sustained release of drugs can improve the healing process and prolong the protectiveness against infection and inflammation [35,36].

Wound repair and regeneration represent a finely tuned phenomenon. The healing process begins with homeostasis, inflammation, tissue remodeling, and growth factors [37,38]. In vivo studies revealed that HPW increased the wound-healing rate and re-epithelization in wounded mice. This could be explained by the fact that HPW treatment stimulated the polarized repair mechanism via the proliferation of the dermal-cell loading of the extracellular matrix and the inhibition of infiltrated cells. Reports confirmed that cellular granulation inhibited the tissue repair and healing process via the activation of granulation markers (hexoseamine and melondialdehyde) [39]. Our results apparently assured that CHDs-HPW reciprocally regulated such markers in the wound site, indicating their effectiveness as a wound-healing approach.

Growth factors play a key role in the wound-healing process. The expression of growth factors attracts fibroblasts to the wound site and produces collagen for tissue repair in the initial period of the wound [39,40]. In this study, HPW upregulated growth factors including VEGF, EGF, and TGF-β. Pro-inflammatory markers IL-1 β and TNF-α were significantly reduced in the CHD-HPW-treated wound. These recognized observations were verified using mRNA and protein growth factors obtained from the wound, improving the load of collagen-I in the wound site. These findings were in general agreement with those reported by Nishida et al. [41], who found that mast cells played an important role in wound healing via the ZnT2/GPR39/IL-6 axis.

The loading of collagen in the wounded site increased epithelization and tissue repair. The results presented in this study demonstrated that the expression of collagen type I was markedly increased in the CHD-HPW-treated group. This indicated that HPW coupled with CHDs increased the loading of collagen from the beginning to after the end of wound healing. HPW alone was unable to cause the significantly high level of collagen in the wound site in mice.

## 4. Materials and Methods

### 4.1. Chemicals, Cell Lines, and Reagents

Collagen hydrolysates were purchased from Sigma Aldrich, and propolis wax purchased from a local market (Al Ahsa, Saudi Arabia). Cell line HACAT was gifted by Dr. Thriugnanasambantham (Pondicherry Centre for Biological Sciences, Pondicherry, India). FTIR was performed by a clinical pharmacy (King Faisal University, Saudi Arabia). All in vitro reagents and chemicals were procured from Invitrogen and Sigma Aldrich (Middle East suppliers).

### 4.2. Propolis Wax Extraction from a Beehive

A total of 100 g of raw propolis (bee hive) was collected in a local market in Al Ahsa, Saudi Arabia, and chopped into small pieces before being soaked in 70% hydroalcohol overnight. The supernatant was collected, and the solvent was removed using a rotary vacuum evaporator. The obtained content, known as propolis extract (HPW), was stored at −20 °C and used for further evaluation studies (Figure 7).

### 4.3. Flavonoids Content Evaluation

HPW hydroalcoholic extract flavonoids were expressed as quercetin equivalents. The calibration curve was created using quercetin (Sigma, Germany) (standard solutions of 5, 10, 20, 40, 60, 80, and 100 mg L1 in 80% ethanol (*V*/*V*)). De Almeida et al. [28] mixed 0.5 mL of PE (ethanolic propolis solution) with the reaction mixture. The absorbance of the reaction mixture was measured at 415 nm after 30 min of incubation at room temperature. Values were expressed as percentages of flavonoids content. The total flavonoids content of the tested samples was calculated using the standard curve’s linear regression equation (y = 32.543x + 0.0607; R^2^ = 0.9974).

#### In Vitro Antioxidant Assay

The antioxidant activity (DPPH radical scavenging activity) of PE was studied in vitro using standard methods. To make the propolis extract solution, we dissolved 100 mg of freeze-dried extract in 10 mL of water and stirred for 1 h to obtain a uniform homogenized latex solution. A volume of 3.9 mL of methanol DPPH solution at 0.6 M/L was added to each diluted solution of PE, ranging from 5 g/mL to 100 g/mL (0.1 mL). After 30 min, the measured absorbance decreased to 517 nm. The percentage of inhibition (PI) (%) was calculated by dividing the percentage of reduced DPPH radical by the percentage of reduced DPPH radical.

It was calculated using the following formula:PI = (Ablank − Asample)/Ablank × 100%,
where Asample is the absorbance of the propolis sample and Ablank is the absorbance of the blank sample.

The experiment was carried out in triplicate. The EC50 value (mg/mL) was calculated for all extracts as the concentration of propolis that caused a 50% decrease in the starting concentration of DPPH radical. The FRAP assay was carried out using the following standard procedure. Moreover, the values were expressed as percentages of PE’s iron chelating activity.

### 4.4. Hemolysis Assay

Blood compatibility was tested using the RBC lysis method. Mouse whole blood was collected, heparinized, and diluted in 0.1 M PBS. Various concentrations of PE were added and incubated for 2 h at 37 °C in an incubator shaker. PBS was used to dilute the sample to 1 mL. Positive and negative controls were blood-treated with 500 μL of Triton X-100 and PBS, respectively. Following incubation, the samples were centrifuged for 10 min at 5000 rpm, and the absorbance of the supernatants was measured at 540 nm.

The following formula was used to calculate the hemolytic percentage:%hemolysis = AS − ANC/APC − ANC × 100
where AS is the absorbance of the test sample, and APC and ANC are the absorbance values of positive and negative controls, respectively.

### 4.5. Fabrication of Collagen Sponge (CHDs) with PE

Collagen hydrolysates were created using the procedure in [26]. In a nutshell, a highly interconnected collagen sponge matrix was created using the gradual freezing method. At 24,000 rpm/10 min/4 °C, 30 mL of 2% *w*/*v* collagen hydrolysate solution in 0.1 M acetic acid was uniformly mixed. Then, using glycol, a well-homogenized collagen solution with 28 mg/cm^2^ PE was incorporated. The mixture was further homogenized for 15 min before being transferred to a freeze-dryer plate and was kept in a freezer for 24 h at constant time intervals of 6, 15, 30, and 78 hr respectively. The frozen samples were then kept in a freeze-dryer at 80 °C under vacuum to obtain a sponge matrix.

### 4.6. In Vitro Drug-Release Study

By placing the HPW-CHD sponge matrix in a protein purification dialysis apparatus, the drug release from the HPW-CHD collagen-loaded scaffold was determined. A magnetic stirrer was used to continuously stir 0.05 M PBS (PBS, pH 7.4) in the receiver compartment [42]. A volume of 1 mL of buffer was tested every 4 h by measuring the absorbance at 220 nm. The PBS buffer was replaced with an equal volume of constant-flow PBS. The absorbance at 218 nm was used to determine the HPW content of the samples (Photo scanner; Thermo Scientific, Waltham, MA, USA). The following equation was used to calculate the percentage of drug released from the HPW-CHD collagen sponge scaffold:A = Qp/Qt × 100
where A is the percentage of drug released from HPW-CH, Qp is the quantity of drug released, and Qt is the total quantity of drug loaded in the biosponge.

### 4.7. In Vitro Enzymatic Degradation

The enzymatic degradation of the collagen sponge matrix was studied according to Muthukumar and colleagues [35]. The collagenase enzyme was used to digest the collagen sponge matrix, which was pre-weighed. In brief, triplicates of each sample (CHDs and CHD + HPW matrix) were taken and dried overnight. The test samples were then incubated for 24 h at 37 °C with 1 mL of 100× concentrated collagenase. Sponge samples were centrifuged, and the dry weight was recorded at 24 h intervals. The weight loss was used to determine the extent of biodegradation of the matrix.

### 4.8. Assay of Cell Migration (Wound-Healing Assay)

In 24-well plates with serum-free DMEM culture medium, the HACAT cell line was seeded (5 × 10^4^ cells/well). Cells were grown until they were 80% confluent. In addition, wells incorporating 0.2 cm^2^ sponge matrix were reseeded. After creating wounds with a sterile 100 μL pipette tip, all wells were washed with medium to remove non-adherent cells. Images were taken with an inverted microscope at various time intervals (0 and 24 h) after sponge incorporation. Optical microscopy (magnification: 200×) was used to monitor cell migration, and the area of the wound-recovered region in each image was calculated using Image J software [43].

### 4.9. Cell Invasion Assay

This assay was carried out in a Matrigel-coated Boyden chamber [44]. In the upper chamber, HACAT cells were grown in culture medium containing CMC/Zn@BTC. The chambers were left at 37 °C overnight. The lower chamber was then filled with 750 μL of DMEM containing 10% FBS. Plates were kept at 37 °C for 36 h. The medium was then removed, and the cells were fixed for a few minutes in 3.7% buffered formalin. Absolute methanol was used to permeabilize the cells for 20 min. For 15 min, the invasive cells were stained with Giemsa stain. The staining was aspirated from the chambers and washed twice with phosphate-buffered saline before counting non-invasive cells with cotton swabs.

### 4.10. In Vivo Studies

All experiments were carried out in accordance with the approval and guidelines of the Institutional Animal Care and Use Committee (IACUC; KFU-REC-2022-MAR-EA000527). Table 1 shows the details of the four groups of male Balb C mice weighing 18 to 22 g. The rats in each group were acclimatized for one week prior to the study and later housed individually under a 12 h light/dark cycle at 25 °C with standard rodent feed obtained from M/s Hindustan Level Ltd. Feeds, Mumbai, India, and water ad libitum.

### 4.11. Dressings and Surgical Procedure

In this study, mice were divided into four groups of five mice each. A subcutaneous injection was used to anesthetize the mice (ketamine at 20 mg/kg of body weight and xylazineat 10 mg/kg of body weight), the dorsal surface of the mice was shaved, and the topical layer was disinfected with Bovidone solution. Surgical procedures were performed as per Muthukumar et al. [42]. Wounds were dressed in sterile cotton gauze and treated with sterile saline alone in the control group (group 1). Group 2 animals received collagen hydrolysates (CHDs); group 3 animals received CHDs-HPW; and group 4 animals received HPW alone. The dressings were changed every three days with the appropriate dressing materials.

Extracted skin wound samples were collected on a regular basis and stored at −80 °C until analysis. The percentages of wound contraction area, as well as immunohistological and biochemical studies, were used to assess the progress of wound healing in mice.

### 4.12. Planimetry: Rate of Contraction and Period of Re-Epithelialization

On the 0th, 5th, 10th, 15th, and 20th days after the wound was created, a digital photograph was taken to document the wound-healing pattern. The time it took for the wound biopsies to fully re-epithelialize was recorded, and the rate of contraction and surface area were measured using the standard planimetric method by tracing the wound on a transparent graph sheet. The following formula was used to calculate the percentage of wound contraction:% of wound contraction = Wound area day 0 − wound area day)/Wound area day 0 × 100
where *n* = number of days (0th day, 5th day, 10th day, 15th day, and 20th day). The data were analyzed using a one-way ANOVA with a 5% margin of error. At the conclusion of the experiments, the tensile strength of the incised wound tissues was measured.

### 4.13. Biochemical Analyses of Excised Wound Tissue

Wound tissues were aseptically excised, and biochemical and cytokine proteins were analyzed using ELISA kits (Invitrogen, City of Carlsbad, CA, USA). The total proteins content in defatted granulation tissue was estimated using the Woessner method, and hexosamine was estimated using the Elson and Morgan method, for both treated and untreated wound tissues.

### 4.14. Real-Time PCR (RT-PCR)

Regenerated skin was collected on the fifth, tenth, and fifteenth days after the wound was created, with all visible fat and healthy skin trimmed for total RNA isolation using a Total RNA extraction kit (Biotechnology, Seoul, South Korea) according to the manufacturer’s instructions [45]. The power cDNA synthesis kit was used to reverse transcribe 300 ng of total RNA (Invitrogen, City of Carlsbad, CA, USA). The following conditions were used for the PCR reaction in a thermal cycler (Applied Biosystems, Foster City, CA, USA): amplification was followed by 5 min at 95 °C and 40 cycles of 15 s at 94 °C, 1 min at 60 °C, with final extension for 5 min at 72 °C min to extend any incomplete single strands. Primer sequences were: VEGF, forward 5′-A GAGTGGGAGGGAAGCTCTTAG3′, reverse 5′-GGGATTTCTTGCGCTTTCG-3′; EGF, forward 5′-TGGAAAAGATGGCTGCCACTGGGTC-3′, reverse 5′-GTGTTCCTCTAGGACCACAAACCA-3′; TGF-β, forward 5′-CACACAGTCCGCTACTTCGT-3′, reverse 5′-CGGG TGCTGTTGTAAAGTGC-3′; and GAPDH forward 5′-CCCACTCCTCCACCTTTGAC-3′, reverse 5′-TGTTGCTGTAGCCAAATTCGT-3′. The relative expression of each gene was calculated using the comparative ΔΔCt calculation technique.

### 4.15. Western Blot

Santa Cruz RIPA lysis buffer was used to lyse excised skin wound tissues (Santa Cruz, Paso Robles, CA, USA). The extracted protein was quantified and loaded into 10% SDS-PAGE with 75 g of protein/well. Polyvinylidene difluoride (PVDF) membranes (pore size of 0.45 m; Bio-Rad, Hercules, CA, USA) were used to transfer the separated proteins [45,46]. Blots were transferred and blocked with 3% BSA before being washed with TBST. Primary antibodies were used to probe blocked blots overnight at 4 °C, according to the manufacturer’s protocol. Primary antibodies Collagen-I and VEGF (both from Invitrogen), and -actin (rabbit polyclonal antibody; 1:2000) (Cell Signaling Technology, Beverly, MA, USA; 4967S) were incubated overnight at 4 °C before being washed with TBST. Washed blots were incubated for 1 h at room temperature with a secondary antibody conjugated with horseradish peroxidase (HRP). The immunoblot bands were visualized using an enhanced chemiluminescence (ECL) system (Pierce, Life Technologies, Austin, TX, USA), and the intensity of the bands was recorded using a LICOR detection system. Image Quant software was used to analyze the expressed bands, and image software v1.8 was used to calculate densitometry.

### 4.16. Statistical Analysis

The results were recorded as means ± SD (*n* = 3). An ANOVA (analysis of variance) and Student’s t-test were used to determine whether there were significant differences among the groups. When a *p* of 0.05 was determined, the observed differences were statistically significant. GraphPad Prism software was used for all statistical analyses.

## 5. Conclusions

In conclusion, the induced accumulation of collagen peptides (CHDs) in the wound microenvironment in mice improved re-epithelization and tissue repair as well as the level of recovery. The combination of CHDs-HPW exhibited a significant reduction in inflammation and inflammatory markers in both in vitro and in vivo experiments. These synergistic combination helped to obtain a faster wound-healing process in a mouse model. Interestingly, HPW upregulated growth factors including VEGF, EGF, and TGF-β and downregulated pro-inflammatory markers such as IL-1 β and TNF-α in wound-treated mice. These responses were highly appreciated in improving the load of collagen-I in the wound site in mice. These results allowed us to conclude that HPW coupled with CHDs increased the loading of collagen from the beginning to after the end of wound healing; therefore, the biosponge architecture could be developed as a promising therapeutic approach for the regeneration and repair of damaged human skin in the future. To the best of the authors’ knowledge, this is the first systematic study showing that CHDs-HPW improved the wound-healing process in an in vivo mouse model via the inactivation of granulation markers (hexoseamine and melondialdehyde).

## Figures and Tables

**Figure 1 molecules-27-07090-f001:**
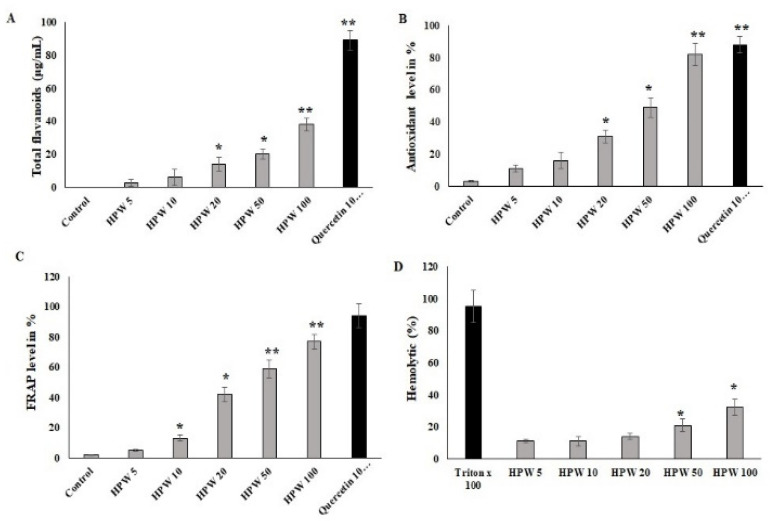
Phytochemical and antioxidant content of honey porpolis was (HPW). (**A**) Total flavonoids content in honey-propolis wax (HPW) and values expressed in μg/mL. (**B**) The percentage of remaining DPPH radical and Fe chelating activity in HPW samples ranged from 5 mg/mL to 100 mg/mL. The antioxidant activity was compared to quercetin antioxidants. The average is the standard deviation (SD) of three independent determinations of each concentration of propolis samples and is shown in the graphs. (**C**) The FRAP assay was performed at high concentrations of HPW, and the results were compared to quercetin. (**D**) HPW had hemolytic activity. Mouse red blood cells were exposed to HPW at concentrations ranging from 5 to 100 mg/mL, and values were determined in triplicate. The results were expressed as means ± SD of triplicate measurements; * *p* < 0.05 and ** *p* < 0.01 when evaluated with control cells.

**Figure 2 molecules-27-07090-f002:**
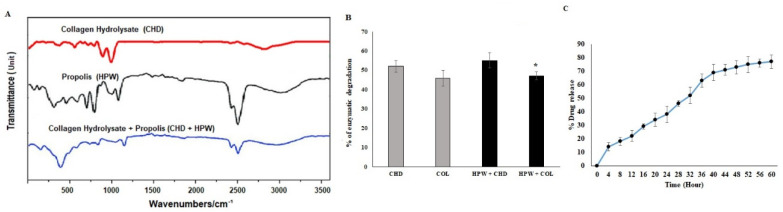
(**A**) FTIR spectra of propolis sponge. Each color represents the specificity of each sample. (**B**) In vitro enzymatic stability of nanofibrous scaffold (* *p* < 0. 05; data presented are means ± SD; *n* = 3). (**C**) In vitro study of release from scaffold. The results were expressed as means ± SD of triplicate measurements; * *p* < 0.05 and *p* < 0.01 when evaluated with control cells.

**Figure 3 molecules-27-07090-f003:**
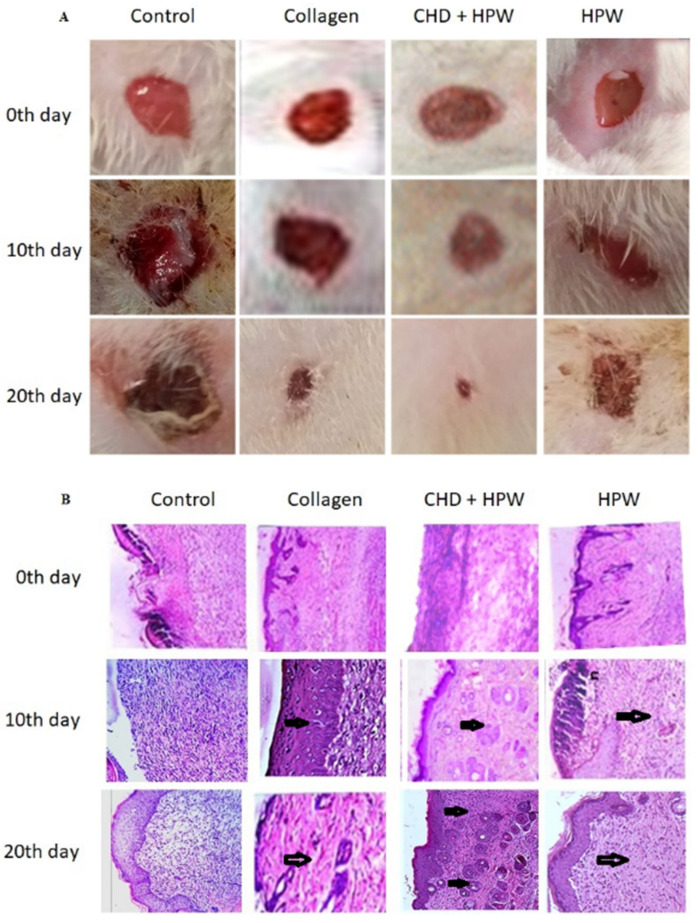
(**A**) Photographic representation of wound-healing pattern on the 0th day, 10th day, and 20th day. (**B**) Microscopic examination of H&E staining of excised skin wounds was performed in different wound groups.

**Figure 4 molecules-27-07090-f004:**
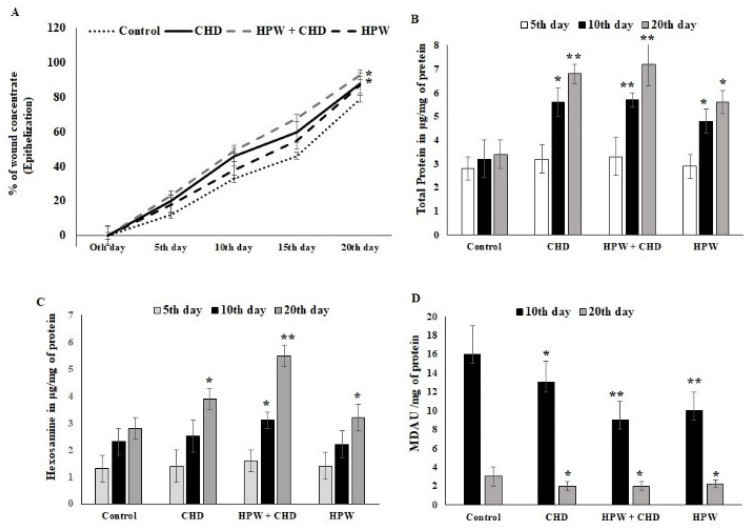
(**A**) Percentage of wound contraction after the 5th, 10th, 15th, and 20th post-wound days. Untreated wounds were used as the control, followed by collagen hydrolysates (CHDs), CHDs-HPW, and HPW. (**B**) Total proteins in excised wound tissue on the 5th, 10th, and 20th post-wound days. (**C**) Hexoseamine in excised wound tissue on the 5th, 10th, and 20th post-wound days, expressed as ug/mg of tissue protein. (**D**) Lipid peroxidation (MDA) in excised wound tissue was measured and expressed as U/mg of protein on the 10th and 20th post-wound days. The results were expressed as means ± SD of triplicate measurements; * *p* < 0.05 and ** *p* < 0.01 when evaluated with control cells.

**Figure 5 molecules-27-07090-f005:**
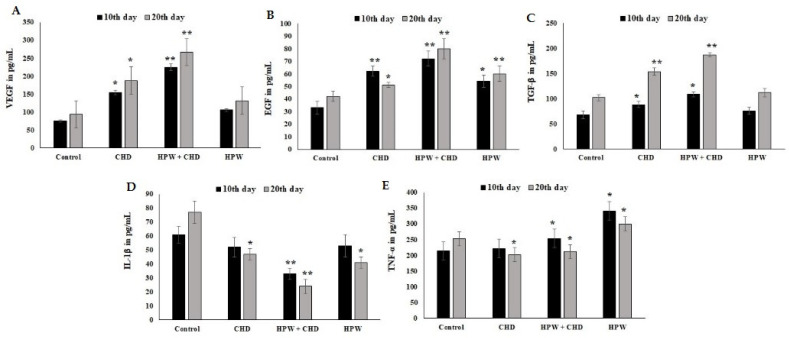
Cytokine protein analyses of excised wound tissue from mice. (**A**) VEGF, (**B**) EGF, (**C**) TGF-Beta, (**D**) IL-1beta, and (**E**) TNF-alpha. The results were expressed as means ± SD of triplicate measurements; * *p* < 0.05 and ** *p* < 0.01 when evaluated with control cells.

**Figure 6 molecules-27-07090-f006:**
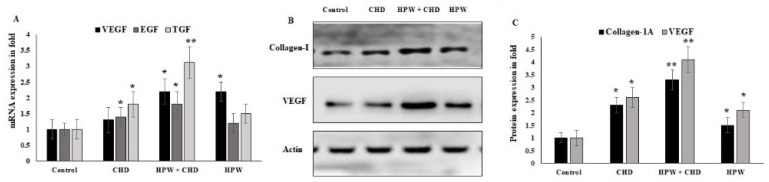
Wound-healing growth factor protein expression in collogen-sponge-based HPW-treated excised wound tissue. (**A**) Real-time PCR was used to quantify VEGF, EGF, and TGF mRNA in excised tissue. As an internal control, GAPDH was used. (**B**) Western blot protein quantification of collagen-I and VEGF protein markers and immunoblot representation. (**C**) Post-wound and under treated conditions, E-cad, ZEB-2, CD-44, SOX-2, and c-M yc were estimated. -actin (actin) was used as an internal control. The results were presented as the means ± SD of a representative experiment from three independent experiments that were studied in quadruplicate and produced similar results. * *p* < 0.05 and ** *p* < 0.01 (one-way ANOVA).

**Figure 7 molecules-27-07090-f007:**
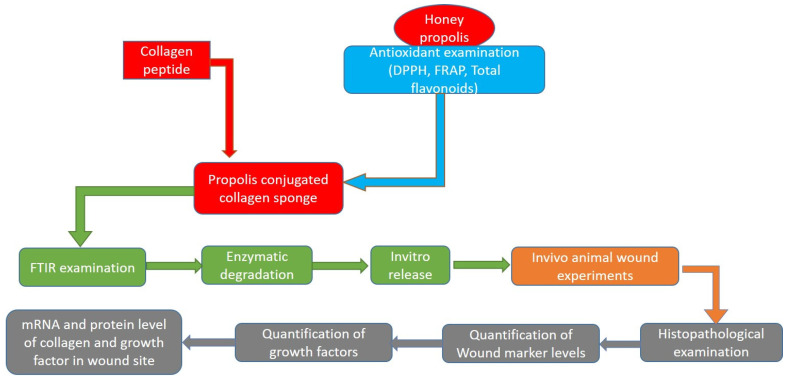
Flow diagram of methodology and experiments performed to achieve the evidence.

**Table 1 molecules-27-07090-t001:** In vivo animal groups and schedule used for each experiment.

Sample Collection	No. of Animals Used for Gross, Biochemical, and Histological Analyses			
	Group 1	Group 2	Group 3	Group 4
	Untreated Control	CHDs	CHD-HPW	HPW
5th day	4	4	5	4
10th day	5	5	5	5
15th day	5	5	5	5
20th day	5	5	6	6

## Data Availability

Data are available from the corresponding author upon request.

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
