# Peer review of "Honey-Propolis-Engineered Collagen Peptides as Promising Wound-Healing Matrix in Mouse Model"

_molecules, 2022, doi:10.3390/molecules27207090_

Round 1
Reviewer 1 Report
Abstract:
Page No 1:
The abstract is very long. The authors simplify the abstract and highlights important points. It will be helpful and easy to understand the researchers of your work.
1. Introduction:
Page No 2:
Line no 44: Worldwide more than six million people struggle with different kinds of wounds.
In this sentence include any example for different kinds of wounds with some recent references.
Page No 2:
Line no 74-76:
Propolis is thought to have antibacterial, anti-inflammatory etc.
Kindly provide related reference to support the current statement.
The authors not included materials and methods section 2. I think, it is included in the result section 4 after discussion.
2. Results:
Page No 4:
Line no 130:
The CHD sponge shows the asymmetric C−O−C bond at 1050 cm−1 , C−O−C glycosidic linkage at 1159 cm−1.
The absorption 1050-1040 cm−1 was commonly CO-O-CO stretching belonging to the class anhydride. Kindly check the current result.
Overall manuscript checks the reference format numbers or name.
In introduction: Sutta et al. investigated the effect of propolis on animal wound healing in clinical and experimental settings [16].
In discussion:
Propolis is a resinous mixture that honey bees produce and contains high phenol and flavonoids (Ramos and Miranda 2007).
Page No 5:
Line no 160:
Figure 2: Improve the quality of picture. Image was not clear.
Improve the discussion part with adding recent scientific evidence of wound healing studies.
5. Conclusion:
Page No 14:
Line no 493:
Improve the conclusion part.
The authors may be added the valuable sentence and important points from the abstract to improve the conclusion part.

Author Response
Authors revised all the comments of the reviewer and appreciate his effort in this manuscript improvement.

Reviewer 2 Report
Authors have studied the Honey propolis engineered collagen peptide for wound healing in the manuscript entitled “Honey propolis engineered collagen peptide as a promising wound healing matrix in mice model”. The manuscript can be well followed but there are some major concerns that I have mentioned section wise. Moreover, English language, punctuations and symbols used in the manuscript also need moderate revisions.
Abstract:
The quantitative data seems to be incomplete in the abstract. Last sentence doesn’t make sense “The newly engineered biosponge could be developed for a rapid skin wound healing in human, in future”. Please reframe it.
Introduction:
Line no. 47: Please correct spacing error. This comment applies throughout the manuscript wherever spacing issue is there in text.
Authors must highlight collagen based previous successful case studies for wound healing. Last para should highlight the novelty of the work and how your study is novel than the previous studies. The main feature of the research study and how it can impact future research should be highlighted in the abstract.
Results and Discussion:
The statistical description in the table and figure must be supplemented as legend/caption wherever applicable. The figures and very blurred and it even hard to read them. It is very difficult to comment on them. It is strongly suggested to give high resolution images so to improve the readability of the manuscript. Figure 3 (A & B): Description should be added by adding arrows to the picture and explaining the particular microscopic changes.
Authors have presented their finding in well manner with appropriate discussions but I still not satisfied with recent works (2020-2022) carried out on similar area. Most of the cited articles in the discussion section are not sufficient. It is suggested to improve the manuscript considering this specific aspect. I wish to see a revised version of the manuscript with a good discussion throughout the result and discussion section with recent. Similarly for other sections the reason of particular findings is not discussed in sufficient way. Also, it is suggested to include a proposed diagram where mechanism of action of collagen-based sponge can be explained in wound healing.
In case of all the figures authors must put some efforts to label the figure so that it become more understandable to the readers.
Section 3: Material and Methods:
Can be well followed but the make of the some of equipment is missing kindly update. Materials need to be explained at first in this section. Make of chemical and other materials used in the study must be included in the manuscript.
Line 338: (DPPH radical-scavenging activity,29)?? I am not able to understand what is 29? Please check such mistakes throughout the manuscript.
Additionally, authors must add a flow diagram showing methodology followed in the experimentation and also show the analysis performed at each stage. This flow diagram will definitely improve the readability of the manuscript.
If authors can take serious efforts to improve the manuscript, I will be happy to re-review the manuscript.
Author Response

(The authors gave the same response as above.)

Reviewer 3 Report
Dear authors,
In many places, the link between section Results and Discussion has disappeared; similarly, the link between section Materials and methods and Results is sorely lacking. This leads to unnecessary repetitions, with non-data-based assertions, distracts the reader from the basic text. This needs to be harmonized to make texts easier for authors to organize and readers to read

Author Response

(The authors gave the same response as above.)

Round 2
Reviewer 2 Report
My previous comment "Additionally, authors must add a flow diagram showing methodology followed in the experimentation and also show the analysis performed at each stage. This flow diagram will definitely improve the readability of the manuscript" is not resolved. I dont know why authors answered as "Author Response: Authors included the methodology flow diagram".
Author Response
Authors included the flow chart of methodology. It was already submitted as graphical abstract about the theme of this study. so authors apologize to the reviewer about this comment misunderstand.
